# Precise electrical gating of the single-molecule Mizoroki-Heck reaction

Lei Zhang [1,2,6], Chen Yang[1,6], Chenxi Lu[3,6], Xingxing Li[4], Yilin Guo[1], Jianning Zhang[2], Jinglong Lin[2], Zhizhou Li[1], Chuancheng Jia[5], Jinlong Yang [4], K. N. Houk [3] ✉, Fanyang Mo [2] ✉ & Xuefeng Guo [1,5] ✉

Precise tuning of chemical reactions with predictable and controllable manners, an ultimate goal chemists desire to achieve, is valuable in the scientific community. This tunability is necessary to understand and regulate chemical transformations at both macroscopic and single-molecule levels to meet demands in potential application scenarios. Herein, we realise accurate tuning of a single-molecule Mizoroki-Heck reaction via applying gate voltages as well as complete deciphering of its detailed intrinsic mechanism by employing an in-situ electrical single-molecule detection, which possesses the capability of single-event tracking. The Mizoroki-Heck reaction can be regulated in different dimensions with a constant catalyst molecule, including the molecular orbital gating of Pd(0) catalyst, the on/off switching of the Mizoroki-Heck reaction, the promotion of its turnover frequency, and the regulation of each elementary reaction within the Mizoroki-Heck catalytic cycle. These results extend the tuning scope of chemical reactions from the macroscopic view to the single-molecule approach, inspiring new insights into designing different strategies or devices to unveil reaction mechanisms and discover novel phenomena.

Precise tuning of chemical reaction outcomes by altering reaction conditions is an area of focus in the chemistry community. It can enable not only desirable reactivity[1], but also spatial and temporal control of chemical processes[2], and is expected to meet the essential and special demands among synthesis[3], materials[4], biotechnology[5], devices[6], three-dimensional (3D) printing[7], etc. In particular, precise control cannot be achieved without an insight into reaction mechanisms, and they are in fact complementary. In combination with the detailed reaction mechanism study, herein we explore the potential of controlling a Mizoroki–Heck reaction with unprecedented precision via gate tuning in a single-molecule transistor.

Gate tuning is a fundamental strategy in the area of (single) molecular electronics[8–11], which has the potential to break the dimension limit of the silicon-based chip technology and bypass Moore's law[12]. In molecular electronic devices, the gate terminal is introduced near organic molecules which are wired between source and drain electrodes[13]. By applying a gate voltage, the frontier molecular orbitals (FMOs) of the wired molecules (bridge molecules), as well as the Fermi energy levels of source and drain electrodes can be tuned precisely[14,15]. For now, single-molecule field-effect transistors (FETs) have been achieved as prototypical devices which employ the redox-active nature of bridge molecules[8,16]. A merit of organic molecules as functional units

[1]Beijing National Laboratory for Molecular Sciences, National Biomedical Imaging Centre, College of Chemistry and Molecular Engineering, Peking University, 292 Chengfu Road, Haidian District, Beijing 100871, P. R. China. [2]School of Materials Science and Engineering, Peking University, Beijing 100871, P. R. China. [3]Department of Chemistry and Biochemistry, University of California, Los Angeles, CA 90095, USA. [4]Hefei National Laboratory for Physical Sciences at Microscale, University of Science and Technology of China, Anhui 230026, P. R. China. [5]Centre of Single-Molecule Sciences, Institute of Modern Optics, Frontiers Science Centre for New Organic Matter, College of Electronic Information and Optical Engineering, Nankai University, 38 Tongyan Road, Jinnan District, Tianjin 300350, P. R. China. [6]These authors contributed equally: Lei Zhang, Chen Yang, Chenxi Lu. ✉e-mail: houk@chem.ucla.edu; fmo@pku.edu.cn; guoxf@pku.edu.cn

in molecular electronics is the wide range of molecule characters for selection, not merely redox. To fully realise the merit of organic molecules and develop new types of single-molecule FETs based on different modes, molecular catalysts have been our focus. We assume that the gate terminal's full and precise tunability would show great potential when applied in catalytic reactions via control of the FMOs of the targeted catalyst molecules. Notably, precise gate-tuning catalysis would have distinct advantages as follows: (I) continuous adjustment of the properties of the bridge catalyst molecules; (II) affecting different reaction processes without changing or modifying the bridge catalyst molecules; and (III) remote and temporal control of catalysis. The introduction of gate-tunable catalysis is a different mode for single-molecule FETs, which provides more choices for molecular electronics and greater possibilities for developing novel devices with special functions. Moreover, once elucidated, the understanding of the mechanism related to gate-tunable catalysis is also instructive for the design of uncommon reactions under electrostatic field[17].

To gain deeper insight into the reaction mechanism and visualise the reaction process, an electrical single-molecule platform, which possesses high electric current resolution and high temporal resolution, has been developed[18]. An advantage of this electrical single-molecule platform is the ability to enable covalent immobilisation of a single molecule and long-term real-time monitoring of the molecular behaviour as well as dynamic disorders under complex reaction conditions. Before conducting the challenging gate-tuning catalysis, we have a certain basis for monitoring the reaction processes. Although it has been realised to discover hidden intermediates/unknown mechanism[19], serve as reversible photoswitch[20], analyse the correlation of electrical spectroscopy and the inventory number of substrates around the single-molecule catalyst[21] regarding reaction dynamics studies, tuning chemical reactions precisely under different stimulations is our long-sought goal. The stimulation, electrical gating, has been employed to research only related to single-molecule transistors in our previous work, which is another basis for exploring gate-tuning catalysis.

In this work, the mechanism of a single-molecule Mizoroki–Heck reaction was deciphered from real-time electrical currents through the reaction centre. Furthermore, machine learning was employed to analyse a large set of experimental data, attempting to realise efficient data processing along with high-accuracy data acquisition in diverse single-molecule platforms. Then, we conducted gate tuning of the Mizoroki–Heck reaction. To what extent gate tuning influenced the catalysis was presented based on mechanistic studies and how the gate voltage resulted in different kinetic processes was discussed using density functional theory (DFT) calculations.

## Results and discussion
### Device fabrication and characterisation
The palladium-catalysed Mizoroki–Heck reaction is widely applied in organic synthesis because of its robust ability to construct C–C bonds. By selecting proper catalyst molecules, the Mizoroki–Heck reaction is robust and insensitive to air and moisture, which shows excellent tolerance to diverse reaction environments. We chose the Mizoroki–Heck reaction as the model reaction to explore gate-tuning catalysis on single-molecule devices. The kernel of the bridge molecule, an N-heterocyclic carbene-palladium (NHC–Pd) complex, is versatile and efficient in cross-coupling reactions[22], as well as the Mizoroki–Heck reaction[23]. We synthesised the bridge molecule and covalently connected it between two graphene point electrodes to form stable Graphene-Molecule-Graphene Single-Molecule Junctions (GMG-SMJs)[24] (Fig. 1a, more synthetic details are presented in Supplementary Note 1 and Supplementary Figs. 1–15). The successful connection of bridge molecules could be determined by the response of current–voltage (I–V) curves (Fig. 1b). With optimised conditions, approximately 17% with ~16 of 92 devices on the same silicon chip

showed responsive current changes alongside voltage variation (Supplementary Fig. 16).

To confirm that only one molecule was connected between the electrodes, the stochastic optical reconstruction microscopy (STORM with single-molecule resolution[25]) effect during the Mizoroki–Heck reaction was investigated. These challenging experiments were conducted on the self-built super high-resolution optical-electrical integrated platform (Supplementary Fig. 17), and realised by using styrene and fluorescent 3-bromoperylene as reaction substrates under basic conditions. The successive Mizoroki–Heck reaction catalysed by the kernel of the GMG-SMJ induced blinking at the catalytic site because of the nonradiative energy/charge transfer from the catalytic centre to graphene electrodes. The resulting single-molecule-resolution fluorescence image (Fig. 1c and Supplementary Fig. 18) showed that only one molecule was connected between the electrodes. Furthermore, the synchronisation of the recorded optical and electrical signal changes exhibited the characteristics of the single-molecule reaction process and further confirmed the single-molecule conjunction (Fig. 1d and Supplementary Movie 1). Then, the single-molecule site was focused and the fluorescence emission spectra were recorded during the reaction. After the addition of styrene to the basic solution of 3-bromoperylene for 5 h, the broadening of the peak with a longer wavelength in the fluorescence emission spectrum was observed (Fig. 1f). In comparison with the macroscopic fluorescence emission spectra of 3-bromoperylene and its cross-coupling product (Fig. 1e), and in further combination with the macroscopic experiment of the Mizoroki–Heck cross-coupling reaction (more details and discussion are presented in Supplementary Notes 1 and 2) and the current level transformation (as discussed below, Fig. 2), we concluded that the connected single molecule can catalyse the Mizoroki–Heck reaction. What is left unclear, however, is how a slight blue shift of the peak with a shorter wavelength occurs whether it is through photobleaching of the cross-coupling product or another associative mechanism.

### Visualisation of the Mizoroki–Heck reaction
To visualise the Mizoroki–Heck reaction process and elucidate a comprehensive mechanism, electrical monitoring with high resolution (nA level for current and ~17 μs for time) was conducted, and the relationship between electrical current levels and the structure of the bridge molecule was elucidated. A N,N-dimethylformamide (DMF) solution of bromobenzene, styrene, and 1,5-diazabicyclo(5,4,0)undec-5-ene (DBU) was added into a home-made reaction cell, which encircled the bridge molecule on the device. After applying a 0.3 V bias voltage between two graphene electrodes, regular periodic current changes along with the time (Fig. 2a, left) were monitored, indicating the progression of the catalytic cycle of the Mizoroki–Heck reaction. The changes in current levels imply the transformation of the structures within the connected single molecule, and the dwell time of a certain current level corresponds to the residence time of a relatively stable intermediate during the reaction. We then utilised machine learning to analyse the number and value of current levels (Supplementary Fig. 19). The values fell into four current ranges (Fig. 2a, right), representing four detectable intermediates during the Mizoroki–Heck reaction. The four current ranges were marked as current levels I, II, III and IV, respectively, from low to high. Figure 2b details an enlarged pattern of multiple current level transformations. To find out the regularity of the transformation, we further utilised programmes to analyse the time trajectories of the four current levels (more details are presented in Supplementary Note 3 and Supplementary Software 1). The transformation processes were presented through statistics starting from current level I (which was attributed to the signal of an NHC–Pd(0) complex vide infra). Figure 2c showed the relationship and reversibility of the current levels (Supplementary Fig. 20 for more details). The results indicated that intermediates related to current levels I and III could be transformed reversibly, as well as III and IV. In

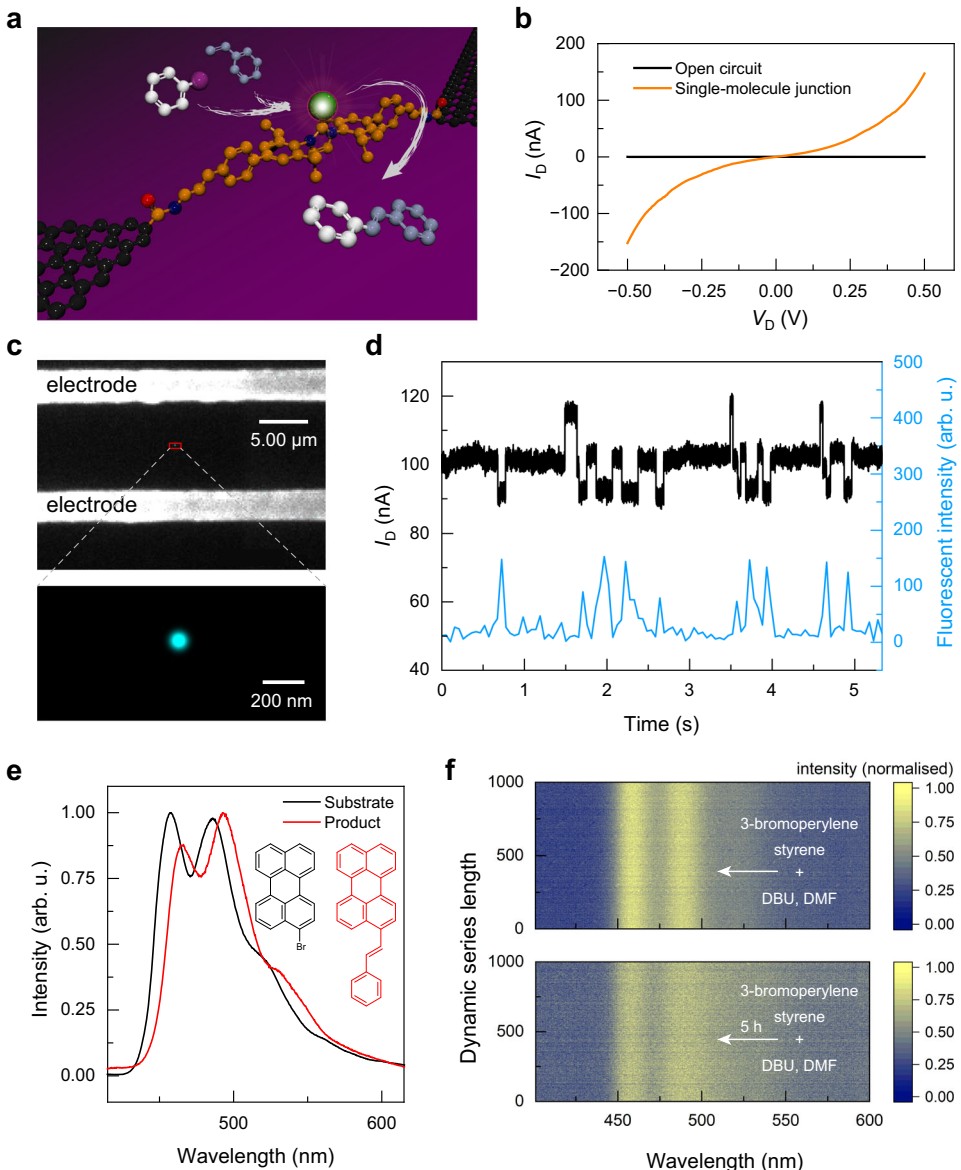

**Fig. 1 | Preparation and characterisation of a single-molecule catalyst device.** **a** Schematic of a single-molecule device for the Mizoroki–Heck reaction. Black ball: carbon atom in graphene; brown ball: carbon atom in the molecular bridge; red ball: oxygen atom in the molecular bridge; dark blue ball: nitrogen atom in the molecular bridge; green ball: palladium atom in the molecular bridge; white ball: carbon atom derived from bromobenzene; purple ball: bromine atom in bromobenzene; grey ball: carbon atom derived from styrene. **b** I–V curves before and after preparation of the single-molecule device. The voltage response indicated the successful preparation of the single-molecule device. **c** Fluorescent super-resolution imaging of the single-molecule catalyst during the Mizoroki–Heck

reaction between 3-bromoperylene and styrene. A 405 nm, 5 mW laser was focused on the graphene device through a ×100 oil lens with 5000 photos taken with an exposure time of 50 ms. **d** The excitation light and 300 mV bias voltages were applied simultaneously at 298 K, and the real-time fluorescent signal of the single-molecule site was compared with the monitored current signal through the single molecule. **e** Normalised fluorescence emission spectra of 3-bromoperylene and its Mizoroki–Heck reaction product in macroscopic experiments. **f** Fluorescent spectroscopies recorded at the single-catalyst reaction site. DBU 1,5-diazabicyclo(5,4,0)undec-5-ene, DMF N,N-dimethylformamide.

contrast, intermediates related to current levels IV and II showed irreversible transformation, as well as II and I. It could be inferred that the electrical Mizoroki–Heck transformation sequence was I→III→IV→II→I periodically. These results were helpful in assigning the structures of different current levels according to the mechanism of the Mizoroki–Heck reaction (Fig. 2d).

Further intermediate-control reactions (Fig. 2e–h and Supplementary Figs. 21–24) and theoretical calculations (Fig. 3a and Supplementary Figs. 25–32) were conducted to support these attributions to assignments. The transformation of the current levels indicated synchronous changes in the GMG-SMJ structure. When the base solution (DBU in DMF) was added to the reaction cell containing a newly

fabricated single NHC–Pd complex, the current dropped to ~6 nA (Fig. 2e, left), which was in line with current level I within the monitored Mizoroki–Heck reaction (Fig. 2e, right and more details in Supplementary Fig. 21). According to previous reports[26], the change after adding DBU should correspond to the pre-activation process and formation of the Pd(0) intermediate. Therefore, current level I should be attributed to the Pd(0) intermediate, and this attribution to the assignment can be further verified by experiments with different halobenzenes (Supplementary Fig. 33). In the following step, the DBU solution was removed and washed, followed by addition of a bromobenzene (PhBr) solution. The current increased to ~22 nA (Fig. 2f, left), which was in line with current level III (Fig. 2f, right and more details in

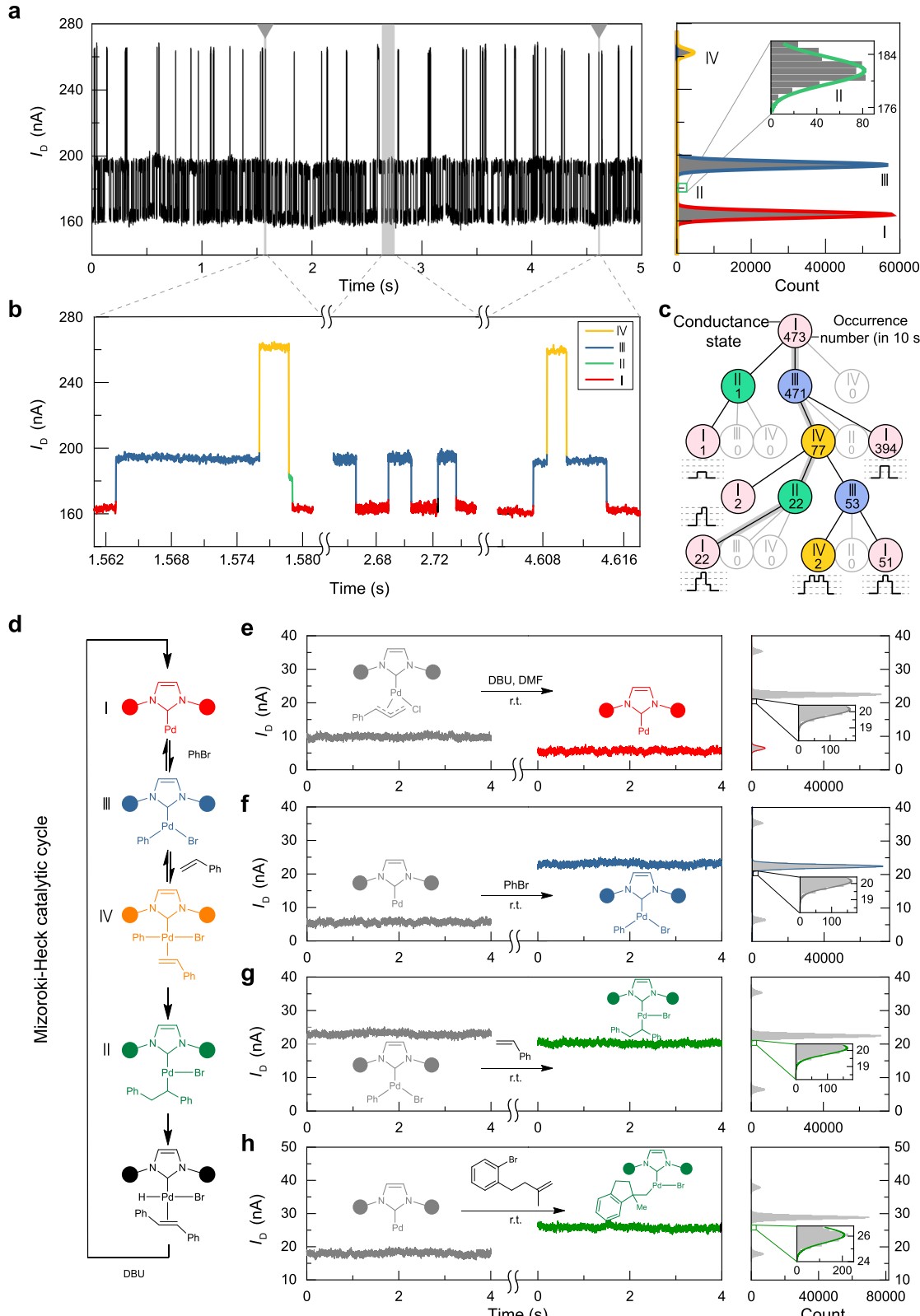

Supplementary Fig. 21), and current level III should be attributed to the oxidative addition intermediate[26]. Subsequently, the styrene solution was added to the cell, and the current dropped slightly to ~20 nA (Fig. 2g, left), which was in line with current level II (Fig. 2g, right and more details in Supplementary Fig. 21). Current level II should correspond to the intermediate after the oxidative addition intermediate within the Mizoroki–Heck catalytic cycle (Fig. 2d). Therefore, current

level II was attributed to the olefin insertion intermediate. This attribution to the assignment can be verified by the reaction between the Pd(0) intermediate and 1-bromo-2-(3-methylbut-3-en-1-yl)benzene (Fig. 2h), and the reaction would stay at the olefin insertion intermediate because of the lack of $\beta$-H[27]. While triethylamine (as reductant) was added, the reductive Mizoroki–Heck reaction of 1-bromo-2-(3-methylbut-3-en-1-yl)benzene would happen[27], and the current level

**Fig. 2 | Electrical monitoring and signal attribution to the assignment of the single-molecule Mizoroki–Heck reaction. a** Current signal variations with a bias voltage of 300 mV between source and drain electrodes in the Mizoroki–Heck reaction conditions at 298 K (left), and corresponding frequency distributions of the current signals (right). Inset: enlarged frequency distribution of the current level II. Reaction conditions: 1 mM PhBr, 1 mM styrene and 1 mM DBU in DMF. **b** Three modes of the current level transformation in **a**. **c** The relationship and reversibility of the current levels via statistics. **d** The Mizoroki–Heck catalytic cycle and the attribution to the assignment of different current levels. **e** The change of the current level after the addition of DBU to the originally connected molecule (left) and frequency distributions of the current signals during the Mizoroki–Heck reaction (right). Inset: enlarged frequency distribution of the current level II. **f** The

change of the current level after the addition of PhBr to the resulting structure from **e** (without DBU, left) and frequency distributions of the current signals during the Mizoroki–Heck reaction (right). Inset: enlarged frequency distribution of the current level II. **g** The change of the current level after the addition of excess styrene to the resulting structure from **f** (left) and frequency distributions of the current signals during the Mizoroki–Heck reaction (right). Inset: enlarged frequency distribution of the current level II. **h** The change of the current level after the addition of 1-bromo-2-(3-methylbut-3-en-1-yl)benzene to the resulting Pd(0) intermediate (another single-molecule device, left) and frequency distributions of the current signals during the reductive Mizoroki–Heck reaction (right). Inset: enlarged frequency distribution of the current level II.

involving the special substrate (Fig. 2h, left) was in line with current level II within the reductive Mizoroki–Heck reaction (Fig. 2h, right and more details in Supplementary Fig. 22). Furthermore, the addition of hydrogen bromide (HBr) and (*E*)−1,2-diphenylethene to the Pd(0) intermediate[28,29] resulted in the current level II, which also supports the attribution to the assignment of current level II (Supplementary Fig. 23). Finally, although the olefin coordination intermediate was not detected via the stepwise-addition reactions, the attribution to the assignment of olefin coordination intermediate can be achieved conveniently by examining the reaction trajectories (Fig. 2b, c). Therefore, current level IV was attributed to the olefin coordination intermediate (Supplementary Fig. 24).

The theoretical calculation results also prove the accuracy of the attributions to assignments. DFT calculations indicate that the intermediates of Pd(0), oxidative addition, olefin coordination and olefin insertion are relatively stable (Fig. 3a and Supplementary Fig. 25). Under basic conditions, the *β*-H elimination intermediate is difficult to dwell to capture because of the low energy barrier and the consumption of HBr by DBU. The highest energy barrier is the reverse process of olefin insertion (considering both forward and reverse processes), which is consistent with the reversibility analysis discussed above (Fig. 2c and Supplementary Fig. 20h). Moreover, these attributions to assignments are consistent with the theoretical simulation of the transmission spectra and *I*–*V* curves (Supplementary Figs. 31 and 32). Collectively, all these results consistently support the attributions to assignments shown in Fig. 2d.

The single-molecule device can tolerate different temperatures, allowing us to study both thermodynamics and kinetics of reactions. We conducted the Mizoroki–Heck reaction at five different temperatures (298, 288, 278, 268 and 258 K), and the signals are presented in Fig. 3b (left) (more temperature-dependent measurements are presented in Supplementary Figs. 34–37). After idealising the *I*–*t* curves (Supplementary Fig. 35), detailed kinetics within Mizoroki–Heck catalytic cycle have been analysed. The dwell times (τ) of different detectable intermediates are longer at decreased temperatures (Fig. 3b, right and Supplementary Fig. 36). In particular, at room temperature (298 K), the dwell times of the four intermediates are $6.1 \pm 0.2$ ms (Pd(0)), $4.8 \pm 0.6$ ms (oxidative addition), $3.0 \pm 0.6$ ms (olefin coordination) and $3.2 \pm 0.4 \times 10^{-1}$ ms (olefin insertion), respectively. The single elementary-reaction rate and its constant can be obtained via $k = r = 1/\langle\tau\rangle$ (approximated as a zero-order reaction at 1 mM according to our previous work[30], Fig. 3c, Supplementary Fig. 37 and Supplementary Tables 1 and 2). The oxidative addition process could be reversible when sterically bulky ligands were introduced[31], and our measurements showed a comparable reversible process in the single-molecule reaction conditions where bulky NHC ligands are applied (Fig. 2b, middle). In addition, the coordination of olefin was also reversible (Fig. 2b, right), which is in agreement with the previous reports[32]. However, the insertion of olefin was irreversible, and it could be attributed to the thermodynamically unfavourable reverse process of olefin insertion (*β*-carbon elimination of the insertion product, Fig. 3a) and the steric hindrance. The transformation between the

olefin insertion intermediate and the Pd(0) intermediate was also irreversible in this reaction system, which involved two main steps, *β*-hydride elimination and reductive elimination/dissociation of the Mizoroki–Heck product. Therefore, the irreversibility could be explained by its non-elementary-reaction process and the low energy barrier of the forward processes, as well as the consumption of HBr by DBU. In brief, this platform enables the investigation of the fundamental process in the catalytic Mizoroki–Heck reaction, especially for olefin insertion and *β*-hydride elimination/product forming steps that are impossible to distinguish by conventional methods[33]. Furthermore, the activation energies of these processes were calculated based on the Arrhenius equation (Fig. 3d). Besides the experimental activation energy values, the kinetic parameters (free energy of activation, enthalpy of activation and entropy of activation) and thermodynamic parameters (free energy change, enthalpy change and entropy change) were also obtained quantitatively (Supplementary Table 3), which are not accessible in previous studies under in situ reaction conditions for the Mizoroki–Heck reaction, demonstrating the capability of our method.

## Gate tuning of the single-molecule Mizoroki–Heck reaction

With the above results in hand, we then investigated single-molecule gate-tuning catalysis. The tuning effect can be visualised or deduced conveniently with the aid of the electrical single-molecule platform. The gate electrode was introduced to the devices, and an ionic liquid (1-butyl-3-methylimidazolium tetrafluoroborate) was employed as the reaction solvent for its ability to deliver an electric field via the formation of an electric double layer (Fig. 4a, b). Different gate voltages were applied with a constant bias voltage ($V_D = 0.3$ V), and the current signals of the Mizoroki–Heck reaction were monitored and recorded (Supplementary Fig. 38).

Current levels representing the intermediates in the Mizoroki–Heck reaction changed as the gate voltage was varied (Supplementary Fig. 39), which manifests that the FMOs of the wired molecules were tuned. The control of FMOs might also be reflected by experimentally observable ionisation energies and electron affinities through Koopmans' Theorem[34]. It was also found that the relative current values regarding the four intermediates under a certain gate voltage remained fixed. To have a clear view of the energy offset between the contact Fermi level and the nearest molecular level responsible for charge transport, the transition voltage ($V_{trans}$) was analysed. When a negative gate voltage was applied, the tunnelling current through GMG-SMJs was enhanced, while a positive gate voltage suppressed the current (Fig. 4c). Next, the Fowler–Nordheim plots of $\ln(I/V^2)$ versus $1/V$ were educed (Fig. 4d), as well as $V_{trans}$ versus $V_G$ for the GMG-SMJ (Fig. 4e). The positive sign of $\alpha = (1.1 \pm 0.0) \times 10^{-1}$ in the GMG-SMJ explicitly indicates that HOMO-mediated tunnelling serves as the dominant transport channel. Furthermore, the molecular orbital shift produced by the applied gate voltage can be also analysed quantitatively via an effective molecular orbital gating energy, $eV_{G,eff} = e |\alpha| V_G$ (Fig. 4f). As the gate voltage reduced from the

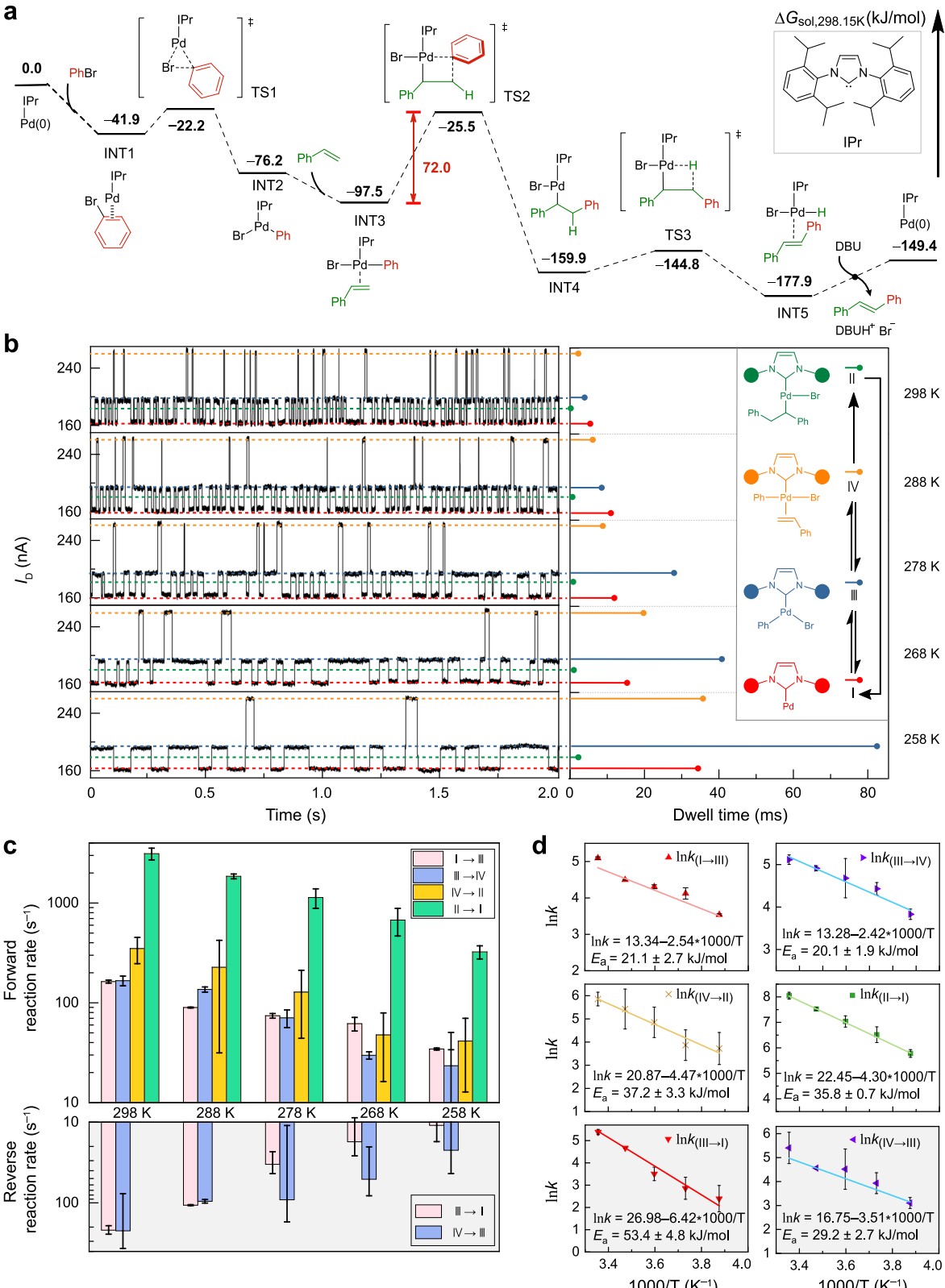

**Fig. 3 | Theoretical free energy surface calculation and experimental kinetics of the single-molecule Mizoroki−Heck reaction. a** Theoretical free energy surface calculation. IPr 1,3-bis(2,6-diisopropylphenyl)−1*H*-imidazol-3-ium-2-ide, TS transition state, INT intermediate. **b** The single-molecule Mizoroki−Heck reaction under different temperatures (left) and the corresponding attribution to the assignment of the four current levels (right). Inset: detected transformations of the current levels. Reaction conditions: 1 mM PhBr, 1 mM styrene and 1 mM DBU in DMF. **c** Rate constants for each elementary reaction at various temperatures. The error scales were derived from the corresponding single exponential fittings of the dwell times (Supplementary Fig. 36). **d** Arrhenius plots of the forward and reverse reactions. The activation energies of the forward and reverse reactions were obtained by fits of the rate constants at the various temperatures using the formula: ln($k$) = ln A − $E_a$/(R × 1000) × (1000/T), A: pre-exponential factor. The error scales were derived from the corresponding single exponential fittings of the dwell times (Supplementary Fig. 36).

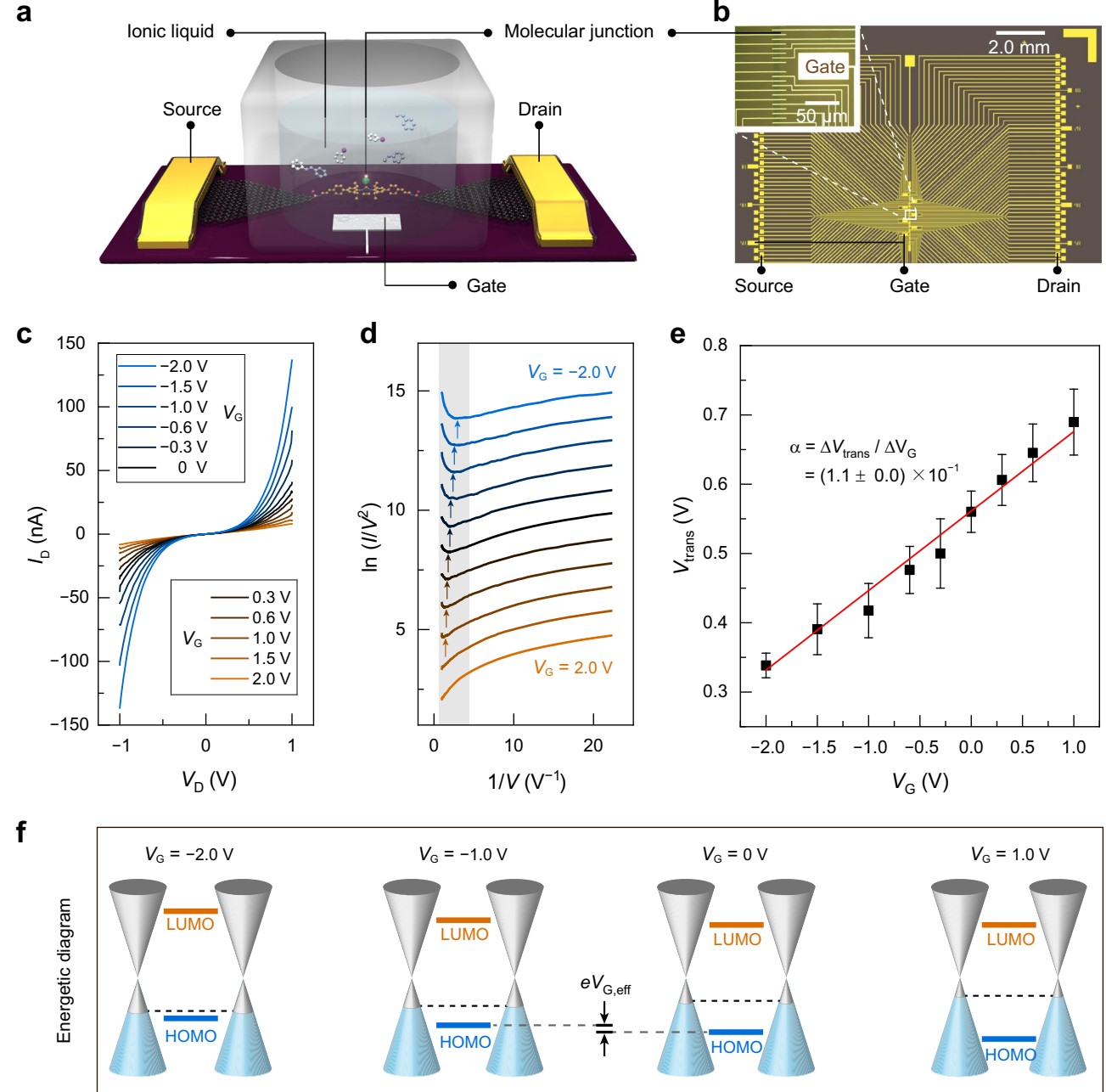

**Fig. 4 | Molecular orbital gating in a single-molecule device. a** Schematic of a single-molecule device with an ionic liquid gate electrode (connected with a remote metal gate electrode) for the Mizoroki−Heck reaction. Ionic liquid: 1-butyl-3-methylimidazolium tetrafluoroborate. **b** Photograph of a real device with metal gate electrodes. Inset: enlarged area of the device. **c** $I-V$ curves measured for different values of $V_G$ based on a Pd(0) GMG-SMJ. **d** Fowler−Nordheim plots corresponding to the $I-V$ curves in **c**. The arrows indicate the boundaries between transport regimes (corresponding to $V_{trans}$). **e** Linear fit of $V_{trans}$ versus $V_G$. $\alpha$: gate efficiency factor. The error scales were derived from the statistics of four different devices. **f** Energetic diagram of the alignment of molecular orbitals relative to the graphene Fermi level in Pd(0) single-molecule transistors under different gate voltages. Effective molecular orbital gating energy, $eV_{G,eff} = e \mid \alpha \mid V_G$. LUMO lowest unoccupied molecular orbital, HOMO highest occupied molecular orbital.

positive to the negative, the HOMO level of the molecular bridge rose energetically and shifted close to the Fermi level. These electrical characteristics of the GMG-SMJ demonstrated molecular orbital gating via our single-molecule platform, which is a harbinger for the useful reaction regulation described below.

Particularly, when the gate voltages are intense enough, the whole Mizoroki−Heck catalytic cycle can be suppressed. At a −2.0 V gate voltage, the intermediate, NHC−Pd(0), could not be monitored, suggesting the blocked catalytic cycle (Fig. 5a, right). At a +2.0 V gate voltage, only the current level of Pd(0) intermediate was observed,

which meant the loss of the catalytic reactivity (Fig. 5b, right). Note that the catalytic process can proceed smoothly without gate voltages (Fig. 2a). Namely, the Mizoroki−Heck reaction can be switched on or off by gate tuning. The electrical single-molecule platform cannot only present the status of a certain reaction, but also provide more details underneath the off status which is usually ignored. The off status under a −2.0 V gate voltage still involves the reaction restricted to transformations among intermediates of oxidative addition, olefin coordination and olefin insertion. This "hidden" information indicates that the unusual reverse reaction between the olefin coordination intermediate

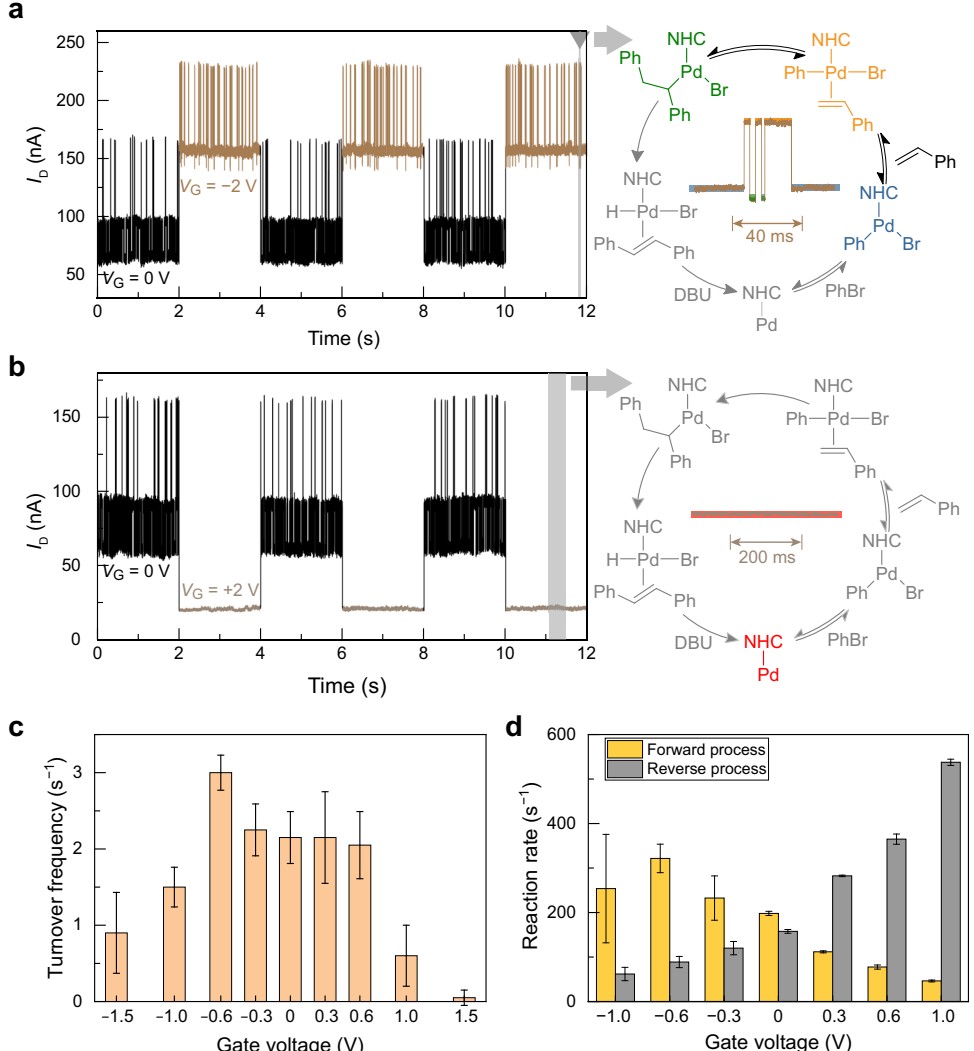

**Fig. 5 | Gate tuning of the single-molecule Mizoroki–Heck reaction.** Reaction conditions: 1 mM PhBr, 1 mM styrene and 1 mM DBU in ionic liquid. **a** Gate-control on/off of the Mizoroki–Heck reaction via applying −2 V/0 V gate voltage (left) and the influence of −2 V gate voltage on the catalytic cycle (right). NHC *N*-heterocyclic carbene. **b** Gate-control on/off of the Mizoroki–Heck reaction via applying +2 V/0 V gate voltage (left) and the influence of +2 V gate voltage on the catalytic cycle (right). **c** The tuning effect of the whole Mizoroki–Heck reaction under different gate voltages. The results were calculated by counting the catalytic cycles in four 5-s intervals (Supplementary Figs. 42–45). The error scales were derived from the statistics of four different devices. **d** The tuning effect of oxidative addition process (reaction between Pd(0) and bromobenzene) and its reverse process. The error scales were derived from the corresponding single exponential fittings of the dwell times (Supplementary Fig. 40).

and the olefin insertion intermediate under gate tuning may provide a complementary avenue for C–C bond activation, which offers a direct approach to editing molecular scaffolds[35,36]. To demonstrate the practicability of gate-tuning on-off of the Mizoroki–Heck reaction, alternative gate voltages of 0 and −2 V (or 0V and +2 V), were applied to the reaction mixture, and instantaneous start or stop of the Mizoroki–Heck reaction was observed (Fig. 5a, b, left). The gate voltages can be applied at any time, which means that precise temporal control of the Mizoroki–Heck reaction is available. Therefore, the electrical gating of single-molecule catalysis has great potential in particular applications, for example, 3D printing (to trigger or terminate a reaction at an arbitrary time). In addition to switching the reaction on and off with freedom, it can be utilised to design new modes of single-molecule FETs and catalytic reactions.

To gain a deeper insight into this electrical gating, the kinetics of the Mizoroki–Heck reaction under different gate voltages were calculated and analysed (Supplementary Figs. 40 and 41). Firstly, the Mizoroki–Heck catalysis can be promoted distinctly under certain gate voltages (Fig. 5c). For example, the

turnover frequency (TOF) can rise to $3.0 \pm 0.2\,\mathrm{s^{-1}}$ under −0.6 V in comparison with the TOF of reaction without gating, $2.2 \pm 0.3\,\mathrm{s^{-1}}$ (the TOF values were calculated based on several sets of gate-voltage-dependent experiments, Supplementary Figs. 42–45). The promotion of a Mizoroki–Heck reaction under gate voltages results from the decrement of the energy barrier of olefin insertion (rate-determining step) within a certain range of gate voltages (Supplementary Figs. 26–28). Secondly, the elementary steps within the catalytic cycle can also be influenced markedly. The oxidative addition process is dominated by the reactivity of the catalyst for certain substrates. More electron-donating ligands on the transition metal catalyst will facilitate the oxidative addition process, while relatively electron-poor catalysts tend to promote the reductive elimination process. The catalyst's reactivity towards oxidative addition can be tuned by electrical fields derived from double layers of ionic liquid when gate voltages are applied. When the negative gate voltages were more intensive, the rate of the oxidative addition process became faster; when the positive gate voltages were bigger, the rate of the

Pd(0) regeneration was faster (Fig. 5d). These regulations originate from the tuning of the FMOs within the NHC–Pd complex via gating, which leads to the different reactivities of the connected catalyst for oxidative addition or reductive elimination (Supplementary Fig. 29). Such a tuning effect elucidates the capacity of the oriented electric field on the modulation of oxidative addition process, which was previously investigated by a detailed computational study[37] and an experimental work based on scanning tunnelling microscope-based break junctions[38]. Furthermore, the gating voltages also influence olefin coordination, olefin insertion, β-H elimination and reductive elimination, and all of the effects on the elementary steps result in the promotion or suppression of the whole Mizoroki–Heck reaction (Fig. 5c and Supplementary Figs. 27–30 and 41). All these kinetic results of the Mizoroki–Heck reaction above present the powerful capability of single-molecule gate tuning, which provides a strategy for tuning the reactivity of catalysts with a constant catalyst molecule and can serve as a model platform to visualise the influence of oriented external electric fields (EEFs). This platform also has the potential to figure out reactions with ambiguous intermediates and discover new reactivities with electrical gating.

In summary, facilitated by high-resolution single-molecule electrical detection, the dynamics of a Mizoroki–Heck reaction have been deciphered, including the visualisation of the reaction trajectory, the detection of hidden intermediates and the quantification of the kinetics of each elementary step within the Mizoroki–Heck reaction. The studies of highly reactive olefin insertion intermediates and related elementary-reaction kinetics under catalytic conditions have been realised, which comprehensively illuminates the Mizoroki–Heck catalytic cycle. More importantly, a gate electrode was introduced to the single-molecule electrical detection platform and unprecedented gate-controllable single-molecule catalysis has been realised. The reactivity for both the whole catalytic cycle and elementary steps can be tuned precisely via electrostatic gating, which presents great potentials in the application of spatially and temporally controllable chemical processes, FET-based single-molecule catalysis and catalysis under oriented EEFs.

## Methods

### Device fabrication and molecular connection

A single layer of high-quality graphene was grown on a 25-μm-thick copper sheet by high-temperature chemical vapour deposition. The graphene was transferred to a 1.5 cm × 1.5 cm silicon wafer with a 300 nm SiO$_2$ layer through PMMA 950. After that, the graphene was protected by a mask and etched by oxygen plasma to obtain a 40-μm wide graphene strip. Then, 8 nm Cr and 60 nm Au layers were thermally evaporated as metal electrode arrays using the template method, with 40 nm SiO$_2$ evaporated onto the metal electrodes to prevent leakage into the solution phase. The prepared graphene transistor was etched with a dashed-line lithography method using electron beam lithography to produce a graphene electrode array with a carboxyl acid terminal using plasma etching and electrical burning. As for devices used for gate-tuning experiments, the gate electrode was patterned by photolithography and electron beam evaporation of 8 nm Cr/60 nm Pt.

We added 10$^{-3}$ mol/L 2-hydroxydiphenylphosphinylbenzene, 1-ethyl-3-(3-dimethylaminopropyl) carbodiimide super-dried CH$_2$Cl$_2$ solution, a catalytic amount of 4-dimethylaminopyridine and i-Pr$_2$NEt to the newly cut graphene device. The reaction was performed for 1.5 days under anhydrous and anaerobic conditions. After that, the device was removed and washed with dry CH$_2$Cl$_2$ and dry THF. This activated the carboxyl group with triphenylphosphine. THF/H$_2$O (10:1) solution with 10$^{-4}$ mol/L of the catalyst was added to the device with the reaction under anaerobic conditions for 1 day. Then, the device was removed, rinsed with THF, and dried with flowing N$_2$. This connected the molecular bridge between the graphene electrode pairs with amide bonds.

### Electrical characterisation

The I–V curve was measured by an Agilent 4155C semiconductor parameter system and a Karl Suss (PM5) manual probe station. The auxiliary output of the HF2LI lock-in amplifier gave a constant bias of 300 mV for the I–t curve. The current signal of the molecular loop was amplified by a DL1211 amplifier and then recorded by a high-speed acquisition card from NIDAQ at a 57600 Sa/s sampling rate. An INSTEC hot and cold chuck (HCC214S, INSTEC) temperature control module with an accuracy of 0.001 °C was placed under the device to regulate the reaction temperature.

### Optical characterisation

The silicon wafer was changed to 1.8 cm × 3.5 cm to ensure the compatibility of the objective lens with the electrical probe system. The probe was connected to the source and drain electrodes of the device and placed on a motorised stage (zdeck). A Nikon Ni-E microscope with a 100× objective lens was positioned in close contact with a ~100-μm-thick home-made polydimethylsiloxane microchannel on the device through the lens oil. The single-molecule device was excited by a 405-nm laser with an EMCCD (Andor) used to receive the feedback emitted light. The STORM process control cable was connected to the trigger terminal of the UHFLI lock-in amplifier to provide synchronous triggering. The reconstruction and analysis of the pictures used the Advanced Research software.

### Theoretical calculation

All the structures were optimised at the B3LYP/6-31G(d)-LANL2DZ[39–42] level of theory. Frequency calculations were performed to verify that intermediates have no imaginary frequency while the transition structures have only one imaginary frequency. Single-point energy calculations were carried out at M06L/6-311++G(2d,p)-SDD[43,44] level of theory. The SMD solvation model[45] of DMF was included in both geometric optimisation and single-point energy calculations for the condition without EEFs. The reported Gibbs free energies were calculated at 298.15 K and 1 M. All the calculations were performed with Gaussian 09 software[46].

### Reporting summary

Further information on research design is available in the Nature Research Reporting Summary linked to this article.

## Data availability

The data that support the findings of this study are available from the corresponding author upon request. Source Data are provided with this paper.

## Code availability

The code used in this study is available in the GitHub database at https://github.com/leizhang-pku/Data-analysis-for-single-molecule-reactions.git and in the Zenodo database at https://doi.org/10.5281/zenodo.6759289.

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

## Acknowledgements

We acknowledge primary financial supports from the National Key R&D Program of China (2017YFA0204901 and 2021YFA1200101), the National Natural Science Foundation of China (22150013, 21727806, 21933001, 21772003 and 22071004), National Science Foundation (CHE-1764328), "Frontiers Science Center for New Organic Matter" at Nankai University (Grant Number 63181206), the Natural Science Foundation of Beijing (2222009) and the Tencent Foundation through the XPLORER PRIZE.

## Author contributions

X.G., F.M. and K.N.H. conceived and designed the experiments. C.Y., Y.G., Z.L, L.Z. and C.J. fabricated the devices and performed the device

measurements. L.Z. and J.Z. carried out the molecular synthesis. C.L., X.L. and J.Y. built and analysed the theoretical model and performed the quantum transport calculation. F.M. and J.L. wrote the machine learning codes for data processing. X.G., F.M., K.N.H., L.Z. and C.Y. analysed the data and wrote the paper. All the authors discussed the results and commented on the manuscript.

## Competing interests

The authors declare no competing interests.
