## [Peer Review File · Nature Communications]

Precise electrical gating of the single-molecule Mizoroki-Heck reactionREVIEWER COMMENTS

Reviewer #1 (Remarks to the Author):

Review Report on, Zhang et al, Ms. NCOM_357798_0
Precise electrical gating of single-molecule catalysis

In this work, Zhang and coworkers studied the Mizoroki-Heck reaction by means of experimental and computational procedures and explored the gate-tuning catalysis on single-molecule devices. They have synthesized a Graphene-Molecule-Graphene Single-Molecule Junctions (GMG-SMJJs) by incorporating covalently bridged N-heterocyclic carbene-palladium (NHC-Pd) complex, which has been utilized for accurately tuning the single-molecule Mizoroki-Heck reaction via electrical gating. Kinetic, thermodynamic, and Turnover frequency values have been determined for this reaction. Furthermore, a comprehensive reaction mechanism is elucidated by electrical monitoring with high resolution. Overall, this study is exciting, and the manuscript is well written. There are some questions below and missing literature citations. These issues should be revised. Then the manuscript can be accepted.

1. Thermodynamic and kinetic study indicates that olefin coordination is reversible, whereas insertion of olefin is irreversible. Is this also confirmed during gate voltage applied to elementary steps of a catalytic cycle? Current level IV is attributed to olefin coordination intermediate. Is gate voltage effect (tuning effect) on coordination of olefin step have observed?
2. The authors quantified the dwell time (τ) for detectable intermediates under different temperature condition. Do you see any changes in the dwell time (τ) for the b-hydride intermediate? Is it possible to quantify the same at lower temperature?
3. The author should cite literature reports where similar work has been conducted: (a) A detailed computational study of the oriented electric field (OEF) effect on the oxidative addition reaction has been reported previously (J. Am. Chem. Soc. 142, 3836-3850 (2020)). The authors have reported that the rate of oxidative addition between palladium catalysts and alkyl/aryl electrophiles can be controlled by an OEF applied along the "reaction axis". Similarly, the current work report fine tuning of NHC-Pd catalyzed reaction with externally applied gate voltage. (b) An experimental work J. Am. Chem. Soc., 2020, 142, 7128, used an STM experiment which showed that an OEF delivered between two gold tips accelerates an oxidative cross coupling reaction between the electron rich Au tip and aryl iodide.
4. Cite the "previous work" corresponding to Fig. 3C.

Reviewer #2 (Remarks to the Author):

The paper describes a combination of a single catalytic center, placed between two graphene electrodes coupled to a single molecule optical microscopy setup.

In the first part of the paper, the methods are introduced, and the system studied using a combination of single molecule electronics, optical microscopy (single molecule level) as well as theoretical modelling. Further, the addition of different reagents are used to verify the local Mizoroki-Heck reaction.

Additionally, the reaction is carried out at different temperatures, verifying the Arrhenius activation behaviour that is expected for the reaction.

In a second series of experiments, the electronic setup is used to "gate" the catalytic center, thus illustrating how catalysis can be turned "on" or "off" using variable gate voltage.

Further, it is studied how different gate voltages can augment the kinetics of the reaction.

In my view this is a very remarkable paper, that not only illustrates a very advanced single molecule platform, but also use this experimental platform to study and gain insights into the function of a catalysis centre at the single molecule level. These insights can be used in future

tailored catalyst design.

In my view, the manuscript is very well written, and the experimental data supports the findings, I recommend publication in its current form.

Reviewer #3 (Remarks to the Author):

Recommendation:
Minor revision

Comments:

In this paper, the authors present a novel method using FET-based devices to control the reactivity of single-molecule catalysis and to track each reaction step within the reaction. The intermediates and kinetic/thermodynamic parameters of the Mizoroki-Heck reaction could be precisely analyzed. Furthermore, the reactivity could be controlled by changing the gate voltage. The concept introduced in this paper is very impressive, and the results are well-organized. Therefore, I recommend the publication of this paper in Nature Communications. Before publication, I suggest the addition of some perspectives about the following issues in the manuscript, which must be informative for potential readers.

- 1) Can this method be applied to reactions in which reaction intermediates are not completely understood?
- 2) Is it possible to observe the product of the Mizoroki-Heck reaction by other tools rather than fluorescent spectroscopy?

Reviewer #4 (Remarks to the Author):

The research area is intriguing and the scientific experiments are fascinating. Reaction tracking at the "single-event level", as stated in the abstract, is absolutely incredible. Indeed, the electric signal from the nanowire is used to produce signals that are interpreted by the authors as single steps of a multistep reaction. The manifold developed by the authors and their choice of target reaction are both good, especially given the numerous potential questions about mechanisms of molecular catalytic reactions that one could project interesting studies onto in the future. The authors cleverly use a beta-hydrogen-resistant substrate to suppress further reaction and assist in assigning one of the reaction intermediates. Figure 5c and 5d are very interesting, presenting data on the effect of applied voltage on turnover frequency of aggregated single-molecular catalysts. The best turnover rate appears to be at zero voltage. Is this observation perhaps because this palladium catalyst has been optimized for zero voltage applications (e.g., "normal use")? It would be interesting to hear the authors' ideas about the context of this observation. The ability to get activation parameters out of individual catalytic steps through the authors' method is very exciting/noteworthy indeed. This data is available in Supplementary Table 3.

In its current form, the manuscript needs work for scientific accuracy and credibility. Many of these points relate to the accuracy of the background literature, but some also relate to the current data and the claims. To address some points, the authors and editor may consider if moving some discussion or experiments from the SI into the manuscript may or may not be beneficial. After addressing the below points, it could be reconsidered for publication.

Scientific Points:

- 1) The manuscript claims the method is "label free" in the abstract and main text. Frankly, this reviewer finds that claim absurd. First, the molecular palladium catalyst is labeled: it is covalently modified and attached with a nanowire. Second, the substrate itself is a fluorescent label.
- 2a) For help in interpreting their microscopic experiments: The authors performed a macroscopic Heck comparison reaction with a similar model catalyst without electrical gating, and similar substrates. This reaction is described in the SI on page 8. Yet, this macroscopic comparison reaction progressed in only 7% yield. The authors state, "The yield of the macroscopic experiment

was relatively low. The main reasons were the unoptimized reactions conditions and side reactions of the model catalyst molecule during pre-activation process." It seems like this catalyst is prone to side reactions. Some may be during the pre-activation process as stated, it seems that some side reactions could be from side reactions with substrates that are nonproductive (e.g., not along the mechanistic reaction pathway). How do the authors know that the signals they are receiving from their microscopic electrical experiments (and analyzing by machine learning) are not also from these side reactions? The signals are assigned only to intermediates along the productive pathways while the catalyst has clear unproductive pathways available to it as seen in the macroscopic reaction. The data could also be interesting if it indicated unproductive pathways, but somehow, there should be a consideration for this possibility. It would be helpful to hear the authors' discussions and considerations about this point before deciding the degree to which the conclusions in the manuscript are justified from the data.

2b) It seems that the very interesting thermodynamic data presented in Supplementary Table S3 may be helpful for answering if the observed states are on-cycle or off-cycle. Specifically, do the values in Table S3 make sense for measure(able) macroscopic parameters of this reaction, as measured through rate of product formation?

3) Figure 1f: after 5 h, the authors claim the aggregated reaction spectrum at single molecules has experienced the expected "red shift" (labeled with red arrow) and assign this shift as consistent with formation of the desired cross-coupled product, on the rational basis of comparison with a measurement of the pure product that is red shifted (Figure 1e). I do see this indicated "red shift" in the provided spectrum after 5 h; however, careful examination shows that there is also a blue shift after 5 h. Specifically, both of the emission peaks are clearly getting *broader*, resulting in apparent shift in both blue and red directions, not just shifting in the red direction. Thus, the provided spectrum is not red shifting in a way that is clearly assignable to the anticipated product, after all. How do the authors account for this apparent spectral broadening process? In what way might it change the data interpretation? It is advised to show both spectra ($t = 0$ and $t = 5$ h) at the same scale of intensity counts (e.g., 287–320 a.u.). This consistency in display may help clarify matters. Consistent display settings are advisable in any case, as apparent shifting could be an artifact of display settings. This comment is not to say that the authors are necessary wrong in their interpretation, only that their currently provided data in Figure 1f does not appear to clearly support their claim.

4) Regarding Figure 1f again: these data show fluorescent spectra (accumulated single-molecules? or perhaps from a single molecule) at the single reaction sites. However, neither the organic starting materials or the organic product is a proposed intermediate that should dwell at the reaction site, on the basis of the authors' own assignment of structures of compounds on the basis of the electrical conduction and DFT studies. How do the authors make sense of this seeming disparity?

5) Figure 5c and 5d. Fascinating data to be able to measure. But there are some questions: Figure 5d shows that higher voltages speed the reverse process. This faster reverse process in 5d is consistent with slower turnover in 5c at the highest voltage 1.0 V. So that makes sense. But, shouldn't the ratio of the forward process rate to the reverse process rate produce a gradual increase in turnover in 5c that matches 5d? What accounts for the apparent inconsistency (a plateau in 5c while 5d gradually changes)? Is the change just below the sensitivity of this technique?

Accurate Context Points:

To some degree, the extent to which scientists praise or enthusiastically justify their own work is a matter of differences in style. Where one must be particularly careful is when the phrasing crosses the line into being scientifically false or unjustified. There are several such statements in this manuscript that should be improved. In fact, much of the writing in the manuscript that does not speak specifically about the data analysis of the current experiments is dubious. The full document should be reread and clarified with this in mind by the authors. Here are a few specific examples:

A) The title oversells. From the generality title, one gets the impression that this paper is the first of its general class, using electrical signals to study single-molecule catalysts broadly. However, the same team of authors published several papers using these single-molecule junctions to study single-molecule organo- and metal catalysts. Prior publications include a paper in 2021 using the

same palladium catalyst with a similar cross-coupling reaction. This prior paper appears not to be cited in the current submission. (Unveiling the full reaction path of the Suzuki–Miyaura cross-coupling in a single-molecule junction. *Nature Nanotechnology* 2021, 16, 1214; <https://www.nature.com/articles/s41565-021-00959-4>). As these prior studies are with voltage, they are example of gating and also of assignment of plausible structures in association with current states. The current manuscript's title should therefore be clarified in order to better reveal and communicate specifically what is new in this manuscript.

B) The abstract claims the technique enables, "conveniently applying gate voltages without modification of the catalyst structure..." This statement is dubious. The authors modify the catalyst structure profoundly. They covalently attach the catalyst to a nanowire, through which different voltages are applied.

C) From the abstract: "This tunability is necessary to understand and regulate chemical transformations at macroscopic and single-molecule levels to meet demands in various application scenarios." It seems implausible (or at least not currently justified by the authors) that tunability at the single-molecule level is a path towards application scenarios, especially given that the sentence before mentions industrial applications. Single-molecule catalysis is a small-scale endeavor in a direct sense.

D) The first two sentences of the introduction are contextually false, demonstrably so. The first two sentences are: "With the rapid evolution of chemistry, the efficiency of chemical transformations is no longer the only goal chemists are pursuing. Precise tuning of chemical reaction outcomes by altering reaction conditions has emerged as an area of focus in the chemistry community." In reality the field of synthetic chemistry has for generations focused on efficiency *with* precise outcome or precise reaction selectivity. Further, precise tuning of chemical reaction outcomes by altering reaction conditions has been a singular focus of synthetic chemistry spanning several generations. This focus area is not emerging. Thus, these introduction sentences are nonsensical. They show a lack of understanding of the literature of synthetic chemistry. With it, there is a lack of understanding/communication of accurate context of the current work and where it might (or might not) fit in beyond the specific research technique and area presented. One alternative angle the authors may wish to consider for context is that often, in synthetic chemistry, the methods available for tuning (e.g., temperature, solvent) affect the reaction overall, rather than be clearly attributable to specific steps. In contrast, with the authors' technique, it may be possible to learn about the effects of voltage tuning on specific reaction steps.

E) Consider removing or judiciously limiting exaggerated writing. One way to do this is to make these phrases more specific and/or less effusive to be accurate. Here are a couple of examples of through the manuscript: "These results clearly extend the tuning scope of chemical..." remove clearly (they couldn't "unclearly" extend, right?). "...novel phenomena at the bottom of nature." What is the bottom of nature? Quarks? "...remarkable... extraordinary" in the same sentence about their research area, remarkable/remarkably used three times in the manuscript, etc.

F) Self-list of advantages of their approach: I, II, and III are somewhat realistic, but catalysis through "(IV) convenient operation via applying electric voltages" is not currently well justified. The authors' do not currently make a plausible case that their or other similar single-molecule covalently modified catalyst nanowire systems is/are "convenient".

G) Small mistake: "thermodynamically unfavorable beta-carbon" could be "beta-hydrogen"

H) The authors' claim to be able to observe "all of the intermediates" and similarly in the conclusion that "the dynamics of a Mizoroki-Heck reaction have been fully deciphered..." The words "all" and "fully" are not justified by the data presented. For example, the likely beta-hydrogen elimination intermediate with olefin coordinated is not observed, nor is the likely pre-oxidative addition intermediate with the arene pi system coordinated (drawn in the DFT calculated pathway).

Listed below are the major changes in the new manuscript:

1. The former title has been revised as “Precise electrical gating of single-molecule Mizoroki-Heck reaction”.

Please see Page 1 in the revised main text and Page S1 in the revised ESI.

2. The first two sentences in the Abstract section and the first sentence at the beginning of the main text have been revised.

Please see Pages 2 and 3 in the in the revised main text.

3. A discussion related to the oriented electric field (OEF) has been added.

Please see Page 13 in the in the revised main text.

4. An addition of our perspectives has been added before the Conclusion Section.

Please see Page 13 in the in the revised main text.

5. We have added the “Code Availability” section after the “Data Availability” section.

Please see Page 16 in the in the revised main text.

6. Figure 1f has been revised.

Please see Page 17 in the in the revised main text.

7. Three references (Refs. 30, 37 and 38) have been added.

Please see Page 26 in the in the revised main text.

8. A statement in the caption of Figure S5 has been added.

Please see Page S26 in the revised ESI.

9. Scheme S8a and its caption have been revised.

Please see Page S32 in the revised ESI.

10. Figure S19 has been added to show the tuning effect on olefin coordination (forward and reverse processes), olefin insertion, β -H elimination and reductive elimination via gating.

Please see Page S47 in the revised ESI.

REVIEWER COMMENTS AND POINT-BY-POINT RESPONSES

Reviewer #1 (Remarks to the Author):

Review Report on, Zhang et al, Ms. NCOM_357798_0
Precise electrical gating of single-molecule catalysis

In this work, Zhang and coworkers studied the Mizoroki-Heck reaction by means of experimental and computational procedures and explored the gate-tuning catalysis on single-molecule devices. They have synthesized a Graphene-Molecule-Graphene Single-Molecule Junctions (GMG-SMJ) by incorporating covalently bridged N-heterocyclic carbene-palladium (NHC-Pd) complex, which has been utilized for accurately tuning the single-molecule Mizoroki-Heck reaction via electrical gating. Kinetic, thermodynamic, and Turnover frequency values have been determined for this reaction. Furthermore, a comprehensive reaction mechanism is elucidated by electrical monitoring with high resolution. Overall, this study is exciting, and the manuscript is well written. There are some questions below and missing literature citations. These issues should be revised. Then the manuscript can be accepted.

General Reply: We thank the reviewer very much for the high evaluation, helpful comments and kind support. All the issues mentioned have been revised, and detailed below are our point-by-point responses according to the reviewer's suggestions. After revisions, the manuscript has been significantly strengthened.

1. Thermodynamic and kinetic study indicates that olefin coordination is reversible, whereas insertion of olefin is irreversible. Is this also confirmed during gate voltage applied to elementary steps of a catalytic cycle?
Current level IV is attributed to olefin coordination intermediate. Is gate voltage effect (tuning effect) on coordination of olefin step have observed?

Our Reply: Thanks a lot for the good comment. We analysed the sequences of current levels during the Mizoroki-Heck reaction in 10 s, and the result was showed as Table R1. When the gate voltage is applied to elementary steps of a catalytic cycle, olefin coordination is still reversible, whereas the reversibility of olefin insertion depends on the magnitude of the gate voltage. If the gate voltage was negative and big enough ($V_G = -2.0$ V or -1.5 V), insertion of olefin is reversible, which is different from the situation without gate voltage. If the gate voltage was in the range from -1.0 to 1.5 V, insertion of olefin is irreversible, which is same as the situation without the gate voltage.

V_G	-2.0 V	-1.5 V	-1.0 V	-0.6 V	-0.3 V	0 V	0.3 V	0.6 V	1.0 V	1.5 V	2.0 V
III→IV	174	324	193	198	196	83	141	129	35	30	0
IV→III	173	301	152	137	125	52	71	57	30	12	0
IV→II	119	44	41	61	71	31	68	72	0	18	0
II→IV	119	21	0	0	0	0	0	0	0	0	0

Table R1 | The reversibility of the current levels (III, IV and II) via statistics in 10 s. When the gate voltage is 2.0 V, only the current level I is observed (Fig. 5b, Figs. S16 and S20–23). When the gate voltage is 1.0 V, the transformation of IV→II can be founded in further monitoring (Figs. S20, 22 and 23). However, the transformation of II→IV is not observed at a 1.0 V gate voltage.

The gate voltage does have the tuning effect on coordination of the olefin step. As shown in Figure R1, the positive gate voltage can accelerate the step, coordination of olefin and its reverse process. Furthermore, the gate voltage has a more prominent influence on the forward process than the reverse process.

Figure R1 | The tuning effect on olefin coordination and its reverse process. For the process III→IV, the dwell time of the current level III is recorded and analysed. The dwell time is shown as the mean value.

Our revision:

(a) We added a statement in the caption of Figure S5 (Page 26 in the Supplementary Information): When the gate voltage is applied to elementary steps of an ongoing catalytic cycle, oxidative addition and olefin coordination are still reversible, whereas the reversibility of olefin insertion depends on the magnitude of the gate voltage (see Scheme S8, Figure S16 and S20–23).

(b) We revised Scheme S8a according to Table R1 (Page 32 in the Supplementary Information).

(c) Figure S19 (Page 47 in the Supplementary Information) was added to show the tuning effect on olefin coordination and its reverse process.

2. The authors quantified the dwell time (τ) for detectable intermediates under different temperature condition. Do you see any changes in the dwell time (τ) for the b-hydride intermediate? Is it possible to quantify the same at lower temperature?

Our Reply: We thank the reviewer very much for the valuable comments and suggestions. During the measurement of the Mizoroki-Heck catalytic cycles in different conditions, we did not detect the β -H elimination intermediate, so we cannot see the changes in the dwell time for the β -H elimination intermediate. Theoretically, the β -H elimination intermediate may be detected in our platform at a lower temperature. We found that the catalytic cycles would be fewer as temperature decreases, and the currents are prone to rest on the signal of oxidative addition intermediate (Fig. 4b). We are considering and developing a higher time-resolution detection to realise the observation of β -H elimination intermediate in our on-going exploration.

3. The author should cite literature reports where similar work has been conducted: (a) A detailed computational study of the oriented electric field (OEF) effect on the oxidative addition reaction has been reported previously (J. Am. Chem. Soc. 142, 3836-3850 (2020)). The authors have reported that the rate of oxidative addition between palladium catalysts and alkyl/aryl electrophiles can be controlled by an OEF applied along the “reaction axis”. Similarly, the current work report fine tuning of NHC–Pd catalyzed reaction with externally applied gate voltage. (b) An experimental work J. Am. Chem. Soc., 2020, 142, 7128, used an STM experiment which showed that an OEF delivered between two gold tips accelerates an oxidative cross coupling reaction between the electron rich Au tip and aryl iodide.

Our Reply: Thanks a lot for the good suggestions. We have studied all these relevant references, and they have been discussed and cited in the manuscript at the proper places.

Our revision:

We added the related discussion (Page 13 in the main text) and corresponding references (Refs. 37 and 38): Such a tuning effect elucidates the capacity of the oriented electric field (OEF) on the modulation of oxidative addition process, which was previously investigated by a detailed computational study³⁷ and an experimental work based on scanning tunneling microscope-based break junctions (STM-BJs)³⁸.

37 Joy, J., Stuyver, T. & Shaik, S. Oriented external electric fields and ionic additives elicit catalysis and mechanistic crossover in oxidative addition reactions. *J. Am. Chem. Soc.* **142**, 3836–3850 (2020).

38 Starr, R. L. *et al.* Gold–carbon contacts from oxidative addition of aryl iodides. *J. Am. Chem. Soc.* **142**, 7128–7133 (2020).

4. Cite the “previous work” corresponding to Fig. 3C.

Our Reply: Thanks a lot for the good suggestions.

Our revision:

The work was cited as Ref. 30 in the main text as follows: Yang, C. *et al.* Unveiling the full reaction path of the Suzuki–Miyaura cross-coupling in a single-molecule junction. *Nat. Nanotechnol.* **16**, 1214–1223 (2021).

Reviewer #2 (Remarks to the Author):

The paper describes a combination of a single catalytic center, placed between two graphene electrodes coupled to a single molecule optical microscopy setup.

In the first part of the paper, the methods are introduced, and the system studied using a combination of single molecule electronics, optical microscopy (single molecule level) as well as theoretical modelling. Further, the addition of different reagents are used to verify the local Mizoroki-Heck reaction.

Additionally, the reaction is carried out at different temperatures, verifying the Arrhenius activation behaviour that is expected for the reaction.

In a second series of experiments, the electronic setup is used to “gate” the catalytic center, thus illustrating how catalysis can be turned “on” or “off” using variable gate voltage.

Further, it is studied how different gate voltages can augment the kinetics of the reaction.

In my view this is a very remarkable paper, that not only illustrates a very advanced single molecule platform, but also use this experimental platform to study and gain insights into the function of a catalysis centre at the single molecule level. These insights can be used in future tailored catalyst design.

In my view, the manuscript is very well written, and the experimental data supports the findings, I recommend publication in its current form.

General Reply: We thank the reviewer very much for the positive remarks, high evaluation, helpful comments and kind support. We are greatly encouraged by the agreement shared by the reviewer about the interest and impact of our work.

Reviewer #3 (Remarks to the Author):

Recommendation:

Minor revision

Comments:

In this paper, the authors present a novel method using FET-based devices to control the reactivity of single-molecule catalysis and to track each reaction step within the reaction. The intermediates and kinetic/thermodynamic parameters of the Mizoroki-Heck reaction could be precisely analyzed. Furthermore, the reactivity could be controlled by changing the gate voltage. The concept introduced in this paper is very impressive, and the results are well-organized. Therefore, I recommend the publication of this paper in Nature Communications. Before publication, I suggest the addition of some perspectives about the following issues in the manuscript, which must be informative for potential readers.

General reply: We thank the reviewer very much for the high evaluation, helpful comments and kind support. Detailed below are our point-by-point responses fully according to the reviewer's suggestions. After revisions, the manuscript has been significantly strengthened.

1) Can this method be applied to reactions in which reaction intermediates are not completely understood?

Our reply: Thanks a lot for the valuable comment. We believe that our platform is capable of investigating reactions with ambiguous intermediates. First of all, we need to hypothesise possible structures of the ambiguous intermediates according to the chemical regulation. Secondly, control reactions and independent synthesis on the platform can facilitate the recognition of certain chemical groups involved. Finally, the theoretical calculation, including density functional theory calculation and transmission spectra calculation, will be conducted to verify whether the hypothesised intermediate structure is reasonable according to thermodynamic/kinetic parameters and relative conductance. Although it is a tough work because of multiple possibilities and extensive efforts, we are searching the interesting reactions and pursuing such explorations.

Our revision:

We added the related statement before *Conclusion* as follows (Page 13 in the main text): This platform also has the potential to figure out reactions with ambiguous intermediates and discover new reactivities with electrical gating.

2) Is it possible to observe the product of the Mizoroki-Heck reaction by other tools rather than fluorescent spectroscopy?

Our reply: Thanks a lot for the valuable comment. The characterisation of the product by NMR or Mass spectra was in our consideration at the beginning. However, we assume that $\sim 10^{-12}$ mol (6.02×10^{11}) molecules can reach the detection limit of NMR or Mass Spectroscopy and the TOF of our single-molecule catalyst is 3 s^{-1} . We can estimate that we need $\frac{6.02 \times 10^{11}}{3} = 2.0 \times 10^{11} \text{ s} \approx 6363 \text{ years}$ to wait for the reaction.

At present, the fluorescent spectroscopy is the available method to observe the product of a single-molecule catalysis. We do contemplate other feasible methods to observe the product.

Reviewer #4 (Remarks to the Author):

The research area is intriguing and the scientific experiments are fascinating. Reaction tracking at the “single-event level”, as stated in the abstract, is absolutely incredible. Indeed, the electric signal from the nanowire is used to produce signals that are interpreted by the authors as single steps of a multistep reaction. The manifold developed by the authors and their choice of target reaction are both good, especially given the numerous potential questions about mechanisms of molecular catalytic reactions that one could project interesting studies onto in the future. The authors cleverly use a beta-hydrogen-resistant substrate to suppress further reaction and assist in assigning one of the reaction intermediates. Figure 5c and 5d are very interesting, presenting data on the effect of applied voltage on turnover frequency of aggregated single-molecular catalysts. The best turnover rate appears to be at zero voltage. Is this observation perhaps because this palladium catalyst has been optimized for zero voltage applications (e.g., “normal use”)? It would be interesting to hear the authors’ ideas about the context of this observation. The ability to get activation parameters out of individual catalytic steps through the authors’ method is very exciting/noteworthy indeed. This data is available in Supplementary Table 3.

General reply: We thank the reviewer very much for the high evaluation, helpful comments and kind support. All the issues mentioned have been revised, and detailed below are our point-by-point responses fully according to the reviewer’s suggestions. After revisions, the manuscript has been significantly strengthened.

The gate voltage influences the Mizoroki-Heck reaction at different dimensions, including turnover frequency (TOF) and reaction rate. As for reaction rate, the catalysis indeed showed high reactivity under zero gate voltage (Fig. 3c). The reversibility of certain steps and the wait for accomplishment of former steps result in an ordinary TOF (Fig. 2c, $\text{TOF} = 22/10 = 2.2$). As for turnover frequency, the catalysis showed the best performance at a -0.6 V gate voltage (Fig. 5c). It derives from the influence of the energy barrier at the rate-determining step (Scheme S6) and the modulations of other elementary reactions (Figure R3 as shown below).

In its current form, the manuscript needs work for scientific accuracy and credibility. Many of these points relate to the accuracy of the background literature, but some also relate to the current data and the claims. To address some points, the authors and editor may consider if moving some discussion or experiments from the SI into the manuscript may or may not be beneficial. After addressing the below points, it could be reconsidered for publication.

Our reply: Thanks a lot for the valuable suggestions. All the concerns raised from the reviewer have been addressed in detail as follows.

Scientific Points:

1) The manuscript claims the method is “label free” in the abstract and main text. Frankly, this reviewer finds that claim absurd. First, the molecular palladium catalyst is labeled: it is covalently modified and attached with a nanowire. Second, the substrate itself is a fluorescent label.

Our reply: Thanks a lot for the valuable comment. According to the suggestion by this reviewer, to avoid ambiguity of the terminology “label-free”, we have removed related words in the manuscript. Here, we also would like to take this opportunity and share our opinion about “label-free”. Typical label-based methods involve detection of fluorescence, chemiluminescence or radioactivity from a specific label. For label-free methods, such label is avoided and some irresponsive chemical groups for certain detection can be used to immobilise the focused objects (for example, *Nat. Nanotechnol.* **6**, 126–132 (2011); *Nat. Protoc.* **1**, 1711–1724 (2006); *Anal. Bioanal. Chem.* **377**, 834–842 (2003)). In the current work, alkyl group was adopted in the catalyst molecule for connection, and it is inert to illumination. During the reaction in order to prove the formation of the product, a fluorescent substrate was particularly used as the reviewer stated. In fact, the monitoring and gating of the catalysis, which is the main body of the manuscript, do not require the fluorescent label.

Our revision: According to the suggestion by this reviewer, to avoid ambiguity of the terminology “label-free”, we have removed related words in the Abstract and Conclusion sections of the main text.

2a) For help in interpreting their microscopic experiments: The authors performed a macroscopic Heck comparison reaction with a similar model catalyst without electrical gating, and similar substrates. This reaction is described in the SI on page 8. Yet, this macroscopic comparison reaction progressed in only 7% yield. The authors state, “The yield of the macroscopic experiment was relatively low. The main reasons were the unoptimized reactions conditions and side reactions of the model catalyst molecule during pre-activation process.”

It seems like this catalyst is prone to side reactions. Some may be during the pre-activation process as stated, it seems that some side reactions could be from side reactions with substrates that are nonproductive (e.g., not along the mechanistic reaction pathway). How do the authors know that the signals they are receiving from their microscopic electrical experiments (and analyzing by machine learning) are not also from these side reactions?

The signals are assigned only to intermediates along the productive pathways while the catalyst has clear unproductive pathways available to it as seen in the macroscopic reaction. The data could also be interesting if it indicated unproductive pathways, but somehow, there should be a consideration for this possibility. It would be helpful to hear the authors’ discussions and considerations about this point before deciding the degree to which the conclusions in the manuscript are justified from the data.

Our reply: Thanks a lot for the valuable comment. Pd(NHC)(cinnamyl)Cl, the catalyst used for the macroscopic Heck comparison reaction, have been investigated adequately in the pre-activation previously. The main side reactions of the catalyst include:

- (a) Formation of inactive Pd(I) dimerisation off-cycle products (*J. Am. Chem. Soc.* **136**, 7300–7316 (2014); *iScience* **23**, 101377 (2020).);
- (b) Formation of Pd black derived from catalyst decomposition (*ACS Catal.* **8**, 3499–3515(2018)).

As mentioned in the literature (*ACS Catal.* **8**, 3499–3515(2018)), “The in situ generation of the Pd(0) highly reactive species can also promote further off-cycle chemistry, including the formation of dimers, trimers, nanoparticles, or Pd-black, which can be detrimental to catalysis”, the issues above resulted in a low yield of macroscopic reaction.

The reasons why we didn’t observe side reactions during single-molecule reactions are summarised as below:

- (a) Only one catalyst molecule was connected between two graphene point electrodes, so the Pd(I)-dimerisation and Pd precipitation cannot happen;
- (b) During the whole reaction process monitored, the substrates are quite excess relative to the single catalyst molecule, so potential side reactions can be suppressed;
- (c) Other potential side reactions, which is irrelevant to the catalyst at the bulk solution, would not influence the monitoring of our focused catalysis.

In addition, according to assignment reactions (Figs. 2e–h) and theoretical calculation (Supplementary Information, section 5), we can confirm that the signals received from microscopic electrical experiments are not derived from these side reactions. Actually, we observed plenty of reverse reaction processes (Fig. 2c and Fig. S5). These processes are unproductive.

2b) It seems that the very interesting thermodynamic data presented in Supplementary Table S3 may be helpful for answering if the observed states are on-cycle or off-cycle. Specifically, do the values in Table S3 make sense for measure(able) macroscopic parameters of this reaction, as measured through rate of product formation?

Our reply: Thanks a lot for the valuable comment. As mentioned above, the observed current level transformations are on-cycle, and the thermodynamic data in Table S3 belong to the Mizoroki-Heck reaction. We have not found macroscopic thermodynamic parameters of this reaction after elaborative search. We believe that the thermodynamic data in Table S3 make sense for the Mizoroki-Heck reaction, especially for the relative magnitudes of different steps. Indeed, further excavation of the information beneath the data is a challenging and long-term goal, which is on-going in our successive studies.

3) Figure 1f: after 5 h, the authors claim the aggregated reaction spectrum at single molecules has experienced the expected “red shift” (labeled with red arrow) and assign this shift as consistent with formation of the desired cross-coupled product, on the rational basis of comparison with a measurement of the pure product that is red shifted (Figure 1e). I do see this indicated “red shift” in the provided spectrum after 5 h; however, careful examination shows that there is also a blue shift after 5 h. Specifically, both of the emission peaks are clearly getting *broader*, resulting in apparent shift in both blue and red directions, not just shifting in the red direction. Thus, the provided spectrum is not red shifting in a way that is clearly assignable to the anticipated product, after all. How do the authors account for this apparent spectral broadening process?

In what way might it change the data interpretation? It is advised to show both spectra ($t = 0$ and $t = 5$ h) at the same scale of intensity counts (e.g., 287–320 a.u.). This consistency in display may help clarify matters. Consistent display settings are advisable in any case, as apparent shifting could be an artifact of display settings. This comment is not to say that the authors are necessary wrong in their interpretation, only that their currently provided data in Figure 1f does not appear to clearly support their claim.

Our reply: Thanks a lot for the high evaluation and valuable comment. We would like to response why we use the different scales of intensity counts firstly. Graphene can adsorb aromatic compounds through π - π stacking (*J. Phys. Chem. Lett.* 2011, 2, 22, 2897–2905; *J. Phys. Chem. Lett.* 2010, 1, 23, 3407–3412). In this experiment, a certain amount of 3-bromoperylene would be adsorbed on graphene within the device. Although the product molecule can also be adsorbed on graphene, it can be detected before moving out from the focused zone *via* diffusion. Therefore, the concentration of 3-bromoperylene decreased after 5 h, which would result in the decrease of the fluorescent intensity. If we use the same scale of intensity counts, only the change of concentration will be reflected (Figure R2a). The relative intensity can reflect the wavelength change within the focused zone. Therefore, we use the different scales of intensity counts. We also have tried to normalise the fluorescent intensity of every single detection. The integrated pictures based on normalisation are shown as Figure R2b.

The spectral broadening process mainly results from the difference of intensity ranges we choose. We have to balance the concentration change and the red shift demonstration. Furthermore, the red shift is more obvious than the blue shift in Fig. 1f, which supports the formation of the desired cross-coupled product.

Figure R2 | Fluorescent spectroscopies recorded at the single-catalyst reaction site on different intensity scales.

Our revision:

Figure 1f have been replaced by Figure R2b (normalised) in Page 17 of the main text.

4) Regarding Figure 1f again: these data show fluorescent spectra (accumulated single-molecules? or perhaps from a single molecule) at the single reaction sites. However, neither the organic starting materials or the organic product is a proposed intermediate that should dwell at the reaction site, on the basis of the authors' own assignment of structures of compounds on the basis of the electrical conduction and DFT studies. How do the authors make sense of this seeming disparity?

Our reply: Thanks a lot for the valuable comment. The fluorescent spectra are derived from a certain area encompassing the single catalyst. The fluorescent signals in this area are recorded 1000 times continuously, and they are integrated to one picture (Fig. 1f). Therefore, the fluorescent spectra are not accumulated single-molecules nor from a

single molecule, and it is from an accumulated small area. Because the organic starting materials and the organic product are not involved in the connected molecular bridge before or after the catalytic cycle, they hardly influence the current signals through the molecular bridge. Therefore, they cannot be detected via electric current and their dwell time is absent.

5) Figure 5c and 5d. Fascinating data to be able to measure. But there are some questions: Figure 5d shows that higher voltages speed the reverse process. This faster reverse process in 5d is consistent with slower turnover in 5c at the highest voltage 1.0 V. So that makes sense. But, shouldn't the ratio of the forward process rate to the reverse process rate produce a gradual increase in turnover in 5c that matches 5d? What accounts for the apparent inconsistency (a plateau in 5c while 5d gradually changes)? Is the change just below the sensitivity of this technique?

Our reply: Thanks a lot for the high evaluation and valuable comment. The change is in the range of the sensitivity for this technique. The turnover frequency is determined not only by the oxidative addition process (forward and reverse processes), but also by other processes, including olefin coordination (forward and reverse processes), olefin insertion, β -H elimination and reductive elimination (Figure R3). The forward olefin coordination (III \rightarrow IV) is sensitive to the gate voltage, and it can be accelerated under more positive gate voltage. The reverse olefin coordination (IV \rightarrow III) has a similar tendency as the forward process, but is less sensitive to the gate voltage. The influence of olefin insertion (IV \rightarrow II) is unobvious under different gate voltages. The combined step (II \rightarrow I), β -H elimination and reductive elimination, is also sensitive to the gate voltage and can be accelerated under a more positive gate voltage. Under a positive gate voltage, the suppression effect of the oxidative addition process is prominent, so higher gate voltages bring a lower turnover frequency. Under a mild negative gate voltage ($-0.6 \sim 0$ V), the acceleration effect of the oxidative addition process is prominent; under an intense negative gate voltage (< -0.6 V), the suppression effect of the olefin coordination and the combined step (β -H elimination and reductive elimination) become prominent.

Figure R3 (Figure S19) | The tuning effect on olefin coordination (forward and reverse processes), olefin insertion, β -H elimination and reductive elimination via gating. For the process III→IV, the dwell time of the current level III is recorded and analysed (the same rule for other processes). The dwell time is shown as the mean value.

Revision:

We added Figure R3 as Figure S19 (Page 47 in the Supplementary Information).

Accurate Context Points:

To some degree, the extent to which scientists praise or enthusiastically justify their own work is a matter of differences in style. Where one must be particularly careful is when the phrasing crosses the line into being scientifically false or unjustified. There are several such statements in this manuscript that should be improved. In fact, much of the writing in the manuscript that does not speak specifically about the data analysis of the current experiments is dubious. The full document should be reread and clarified with this in mind by the authors. Here are a few specific examples:

General reply: Thanks a lot for the valuable suggestions. All the concerns raised from the reviewer have been addressed in detail as follows and revised in the main text.

A) The title oversells. From the generality title, one gets the impression that this paper is the first of its general class, using electrical signals to study single-molecule catalysts broadly. However, the same team of authors published several papers using these single-molecule junctions to study single-molecule organo- and metal catalysts. Prior publications include a paper in 2021 using the same palladium catalyst with a similar cross-coupling reaction. This prior paper appears not to be cited in the current submission. (Unveiling the full reaction path of the Suzuki–Miyaura cross-coupling in a single-molecule junction. *Nature Nanotechnology* 2021, 16, 1214; <https://www.nature.com/articles/s41565-021-00959-4>). As these prior studies are with voltage, they are example of gating and also of assignment of plausible structures in association with current states. The current manuscript's title should therefore be clarified in order to better reveal and communicate specifically what is new in this manuscript.

Our reply: Thanks a lot for the valuable suggestions. We have revised the title as “Precise electrical gating of single-molecule Mizoroki-Heck reaction”. In our prior works, the gate voltage has not been applied to tune the reaction (the voltage applied is bias voltage between source and drain electrodes). It is the first time to present an electrical gating catalysis based on Graphene-Molecule-Graphene Single-Molecule Junctions (GMG-SMJs).

Our revision: To make the title more specific, we have revised it accordingly as follows: “Precise electrical gating of single-molecule Mizoroki-Heck reaction”.

B) The abstract claims the technique enables, “conveniently applying gate voltages without modification of the catalyst structure...” This statement is dubious. The authors modify the catalyst structure profoundly. They covalently attach the catalyst to a nanowire, through which different voltages are applied.

Our reply: Thanks a lot for the valuable suggestions. We have revised the sentence as “conveniently applying gate voltages with a constant catalyst molecule...”.

Modification of the catalyst structure is a popular method to tune the macroscopic reactions. We aim to show that the tuning effect could be realised without modification of the catalyst structure after the connection. To avoid the potential ambiguity, we have revised it.

Our revision: To avoid the potential ambiguity, we have revised the sentences in Pages 2 and 13 of the main text accordingly as follows: “with a constant catalyst molecule”.

C) From the abstract: “This tunability is necessary to understand and regulate chemical transformations at macroscopic and single-molecule levels to meet demands in various application scenarios.” It seems implausible (or at least not currently justified by the authors) that tunability at the single-molecule level is a path towards application scenarios, especially given that the sentence before mentions industrial applications. Single-molecule catalysis is a small-scale endeavor in a direct sense.

Our reply: Thanks a lot for the valuable suggestions. We have revised the sentences as “...is valuable in the scientific community” and “This tunability is necessary to understand and regulate chemical transformations at both macroscopic and single-molecule levels to meet demands in potential application scenarios.” Here, we would like to share our opinions about the application of electrical gating of single-molecule catalysis:

- (a) It can be used directly to investigate reaction mechanism and the influence of gate voltages;
- (b) The tuning effect of gate voltages can facilitate the design of new reaction under oriented electric fields (OEFs);
- (c) It offers inspiration to design new modes of molecular field-effect transistors (experimentally exhibited in Figure S17) that combine chemical reactivity and electrical functions, and single-molecule devices show a fascinating application in new generation integrated chip;
- (d) Integration of Graphene-Molecule-Graphene Single-Molecule Junctions (GMG-SMJ) is our on-going research interest, and the integration may achieve controllable synthesis at the macroscopic level.

Our revision: We have revised the sentences as follows (Page 2 in the main text): “...is valuable in the scientific community” and “This tunability is necessary to understand and regulate chemical transformations at both macroscopic and single-molecule levels to meet demands in potential application scenarios.”

D) The first two sentences of the introduction are contextually false, demonstrably so. The first two sentences are: “With the rapid evolution of chemistry, the efficiency of chemical transformations is no longer the only goal chemists are pursuing. Precise tuning of chemical reaction outcomes by altering reaction conditions has emerged as an area of focus in the chemistry community.” In reality the field of synthetic chemistry has for generations focused on efficiency *with* precise outcome or precise reaction

selectivity. Further, precise tuning of chemical reaction outcomes by altering reaction conditions has been a singular focus of synthetic chemistry spanning several generations. This focus area is not emerging. Thus, these introduction sentences are nonsensical. They show a lack of understanding of the literature of synthetic chemistry. With it, there is a lack of understanding/communication of accurate context of the current work and where it might (or might not) fit in beyond the specific research technique and area presented. One alternative angle the authors may wish to consider for context is that often, in synthetic chemistry, the methods available for tuning (e.g., temperature, solvent) affect the reaction overall, rather than be clearly attributable to specific steps. In contrast, with the authors' technique, it may be possible to learn about the effects of voltage tuning on specific reaction steps.

Our reply: Thanks a lot for the valuable suggestions. We have revised the first two sentences as “Precise tuning of chemical reaction outcomes by altering reaction conditions is an area of focus in the chemistry community.” Indeed, tuning specific steps within Mizoroki-Heck reaction is only one feature of our work, and we would like to start the manuscript from a relative broad angle. Thank the reviewer very much again for the constructive suggestions.

Our revision: We have revised the first two sentences (Page 3 in the main text) as “Precise tuning of chemical reaction outcomes by altering reaction conditions is an area of focus in the chemistry community.”

E) Consider removing or judiciously limiting exaggerated writing. One way to do this is to make these phrases more specific and/or less effusive to be accurate. Here are a couple of examples of through the manuscript: “These results clearly extend the tuning scope of chemical...” remove clearly (they couldn't “unclearly” extend, right?). “...novel phenomena at the bottom of nature.” What is the bottom of nature? Quarks? “...remarkable... extraordinary” in the same sentence about their research area, remarkable/remarkably used three times in the manuscript, etc.

Our reply: Thanks a lot for the valuable suggestions.

Our revision: We have removed related words (clearly, remarkable, extraordinarily and so on) and revised the whole manuscript fully according to the reviewer's suggestion.

F) Self-list of advantages of their approach: I, II, and III are somewhat realistic, but catalysis through “(IV) convenient operation via applying electric voltages” is not currently well justified. The authors' do not currently make a plausible case that their or other similar single-molecule covalently modified catalyst nanowire systems is/are “convenient”.

Our reply: Thanks a lot for the valuable suggestions.

Our revision: We have deleted the sentence accordingly.

G) Small mistake: “thermodynamically unfavorable beta-carbon” could be “beta-hydrogen”

Our reply: Thanks a lot for the valuable suggestions. This expression is in accordance with our theoretical calculation in Fig. 3a. The energy barrier for β -carbon elimination is high (INT4 \rightarrow INT3), which results in the irreversible process. To make the expression more clear, we have revised the sentence as “...it could be attributed to the thermodynamically unfavorable reverse process of olefin insertion (β -carbon elimination of the insertion product)...”.

Our revision: To make the expression more clear, we have revised the sentence as follows (Page 9 in the main text): “...it could be attributed to the thermodynamically unfavorable reverse process of olefin insertion (β -carbon elimination of the insertion product)...”.

H) The authors’ claim to be able to observe “all of the intermediates” and similarly in the conclusion that “the dynamics of a Mizoroki-Heck reaction have been fully deciphered...” The words “all” and “fully” are not justified by the data presented. For example, the likely beta-hydrogen elimination intermediate with olefin coordinated is not observed, nor is the likely pre-oxidative addition intermediate with the arene pi system coordinated (drawn in the DFT calculated pathway).

Our reply: Thanks a lot for the valuable suggestions.

Our revision: We have removed related words (fully, full and all) and revised the sentence related to “all of the intermediates” as follows (Page 11 in the main text): “...the intermediate, NHC–Pd(0), could not be monitored”.

Finally, we would like to thank all the referees very much for their patience, precious time and kind support.

REVIEWER COMMENTS

Reviewer #4 (Remarks to the Author):

The revised manuscript presents exciting science. The authors have addressed almost all of my comments satisfactorily. Regarding revised Figure 1f, I maintain that the spectrum after 5 h does not support the authors' claim of red shift. There are two peaks in the spectrum. Examine, for example, the peak between 450-470 nm. This peak appears to *blue shift*, the opposite of the authors' conclusion. The other peak, at ~475-500 nm, simply appears to broaden (per my original comment to the authors). I still do not see that the data supports a red shift upon which the authors are building their argument. A red shift should occur to both peaks if the authors' interpretation is correct. I thus urge the authors to reconsider this aspect of their data interpretation prior to publication.

Listed below are the major changes in the new manuscript:

1. The discussion related to the fluorescence spectroscopies recorded at the single-catalyst reaction site (Figure 1f) has been revised.
Please see Page 6 in the revised main text.

Former version: After addition of styrene to the basic solution of 3-bromoperylene for 5 h, a red shift of the fluorescence emission spectra was observed (Fig. 1f). In comparison with the macroscopic fluorescence emission spectra of 3-bromoperylene and its cross-coupling product (Fig. 1e), we concluded that the connected single molecule can catalyse the Mizoroki-Heck reaction.

Revised version: After addition of styrene to the basic solution of 3-bromoperylene for 5 h, the broadening of the peak with longer wavelength in the fluorescence emission spectrum was observed (Fig. 1f). In comparison with the macroscopic fluorescence emission spectra of 3-bromoperylene and its cross-coupling product (Fig. 1e), and in further combination with the macroscopic experiment of the Mizoroki-Heck cross-coupling reaction (more details and discussion are presented in the Supplementary Information) and the current level transformation (as discussed below, Fig. 2), we concluded that the connected single molecule can catalyse the Mizoroki-Heck reaction. What is left unclear, however, is how a slight blue shift of the peak with shorter wavelength occurs whether it is through photobleaching of the cross-coupling product or another associative mechanism.

2. The annotation of Figure 1f has been revised.
Please see Page 17 in the revised main text.

Former version: There are the words, “red shift”, and one shift-direction arrow in Figure 1f.

Revised version: The words, “red shift”, and the arrow in Figure 1f have been removed.

REVIEWER COMMENTS AND POINT-BY-POINT RESPONSES

Reviewer #4 (Remarks to the Author):

The revised manuscript presents exciting science. The authors have addressed almost all of my comments satisfactorily. Regarding revised Figure 1f, I maintain that the spectrum after 5 h does not support the authors' claim of red shift. There are two peaks in the spectrum. Examine, for example, the peak between 450–470 nm. This peak appears to *blue shift*, the opposite of the authors' conclusion. The other peak, at ~475–500 nm, simply appears to broaden (per my original comment to the authors). I still do not see that the data supports a red shift upon which the authors are building their argument. A red shift should occur to both peaks if the authors' interpretation is correct. I thus urge the authors to reconsider this aspect of their data interpretation prior to publication.

Our Reply: We thank the reviewer very much for his/her helpful comments and kind support. Fully according to the reviewer's suggestions, the data interpretation has been revised. Detailed below are our point-by-point responses. After revisions, the manuscript has been significantly strengthened.

After examining the data and Figure 1f carefully, we agree with the reviewer's comment. The change of the fluorescent peak might originate from different factors (for example, photoreduction of bromide or photobleaching of the starting material and cross-coupling product), which need further studies. In the current case, in combination with the broadening of the peak with longer wavelength (~475–500 nm, Fig. 1f), the macroscopic experiment of the Mizoroki-Heck cross-coupling reaction (Please see the Supplementary Information) and the current level transformation (Fig. 2), we can conclude that the connected single molecule can catalyse the Mizoroki-Heck reaction. To make the manuscript more accurate according to the reviewer's suggestions, we delete the statement of "red shift".

Our revision:

(a) The discussion related to the fluorescence spectroscopies recorded at the single-catalyst reaction site has been revised as follows (Page 6 in the main text):

After addition of styrene to the basic solution of 3-bromoperylene for 5 h, the broadening of the peak with longer wavelength in the fluorescence emission spectrum was observed (Fig. 1f). In comparison with the macroscopic fluorescence emission spectra of 3-bromoperylene and its cross-coupling product (Fig. 1e), and in further combination with the macroscopic experiment of the Mizoroki-Heck cross-coupling reaction (more details and discussion are presented in the Supplementary Information) and the current level transformation (as discussed below, Fig. 2), we concluded that the connected single molecule can catalyse the Mizoroki-Heck reaction. What is left unclear, however, is how a slight blue shift of the peak with shorter wavelength occurs whether it is through photobleaching of

the cross-coupling product or another associative mechanism.

(b) The annotation of Figure 1f has been revised. The words, “red shift”, and the shift-direction arrow in Figure 1f have been removed shown as below (Also please see Page 17 in the main text).